# Oxidative stress and associated clinical manifestations in malaria and sickle cell (HbSS) comorbidity

**Enoch Aninagyei**[1], **Clement Okraku Tettey**[1], **Henrietta Kwansa-Bentum**[1], **Adjoa Agyemang Boakye**[1], **George Ghartey-Kwansah**[2], **Alex Boye**[3], **Desmond Omane Acheampong**[2]*

1 School of Basic and Biomedical Sciences, Department of Biomedical Sciences, University of Health and Allied Sciences, Ho, Ghana, 2 Department of Biomedical Sciences, School of Allied Health Sciences, University of Cape Coast, Cape Coast, Ghana, 3 Department of Medical Laboratory Science, School of Allied Health Sciences, University of Cape Coast, Cape Coast, Ghana

* dacheampong@ucc.edu.gh

## Abstract

In Ghana, uncomplicated malaria and sickle cell disease (SCD) is common, hence comorbidity is not farfetched. However, the extent of oxidative stress and the array of clinical manifestations in this comorbidity (presence of both malaria and SCD) has not been fully explored. This study highlights the impact of uncomplicated malaria on SCD. The level of isoprostane, 8-iso-prostaglandin F2α (8-iso-PGF2α) was used to assess oxidative stress while plasma biochemistry and urinalysis was used to assess renal function. Hematological profiling was also done to assess the impact of comorbidity on the hematological cell lines. Of the 411 study participants with malaria, 45 (11%) had SCD. Mean body temperature was significantly higher in comorbidity compared to malaria and SCD cohorts, while a lower parasite density range was obtained in comorbidity compared to malaria cohorts. Furthermore, in comorbidity, the 8-iso-PGF2α oxidative stress biomarker was significantly elevated in all ages, parasite density ranges and gender groups. Comorbidity affected both leukocytic and erythrocytic cell lines with significant eosinophilia and monocytosis coexisting with erythrocytic parameters consistent with severe anemia. Biochemically, while plasma creatinine and bilirubin were significantly elevated in comorbidity, spot urinary creatinine was significantly reduced. Additionally, urine samples in the comorbid state were slightly acidic and hypersthenuric with significant hematuria, proteinuria, and bilirubinemia. Finally, 80% or more malaria-SCD presented with chills, fever, anorexia, headache, joint pains, lethargy, and vomiting. In conclusion, malaria could induce vaso-occlusive crisis in sickle cell disease, therefore, prompt management will alleviate the severity of this comorbidity.

## Introduction

Uncomplicated malaria is common in Ghana, especially in the Greater Accra region, albeit, at low prevalence [1, 2]. On the other hand, sickle disease is also prevalent in Ghana with about

---

**Data Availability Statement:** All data are contained within the paper and its Supporting information files.

**Funding:** The authors received no specific funding for this work. The study was funded from authors' resources. Prof. Desmond Omane Acheampong provided personal funds for the oxidative stress biomarker while Dr Enoch Aninagyei also provided personal funds for the haematological, renal assays and the urinalysis.

**Competing interests:** The authors have declared that no competing interests exist.

**Abbreviations:** EBR, eosinophils-to-basophils ratio; EMR, eosinophils-to-monocytes ratio; HbS, Sickle cell hemoglobin; HbSS, two HbS haplotypes; LBR, lymphocytes-to-basophils ratio; LER, lymphocytes-to-eosinophils ratio; LMR, lymphocytes-to-monocytes ratio (LMR); malaria-SCD, malaria in sickle cell disease; MBR, monocytes-to-basophils ratio; MCH, mean cell hemoglobin; MCV, mean cell volume; NBR, neutrophils-to-basophils ratio; NER, neutrophils-to-eosinophils ratio; NLR, neutrophils-to-lymphocytes ratio; NMR, neutrophils-to-monocytes ratio; SCD, Sickle cell disease.

5000 births per year [3]. Between 2013 and 2014, 5,451 patients with sickle cell hemoglobin visited the sickle cell clinic at Korle-Bu Teaching Hospital in Accra, Ghana, for medical attention. Of this number, 55% had homozygous hemoglobin S (HbSS) [4]. Hemoglobin S (HbS) results from the substitution of hydrophilic glutamic acid by hydrophobic valine at position six in the β-globin chain [5]. Globally, 3.2 million people live with HbSS or HbSC. About 176,000 people die of HbS disease related complications every year [6]. Anaemia is common in sickle patients [7] together with vaso-occlusion which frequently leads to ischemia. These cascade of events are the predominant pathophysiology responsible for acute painful vaso-occlusive crisis (VOC) [8]. Increased plasma viscosity occurs as a result of chronic hemolysis and reduced sickle red cell deformability due to HbS polymerization [9]. These effects could be prevented or reversed by therapies that prevent HbS polymerization by allosterically modifying HbS oxygen affinity, preventing erythrocyte dehydration. Hydroxyurea, metformin and sodium butyrate are common examples [10]. Sickle cell patients are generally stable unless there is an onset of sickle cell crisis. Sickle cell crisis, presented as aplastic crisis, splenic sequestration crisis, hyperhemolytic crisis, hepatic crisis, dactylitis, and acute chest syndrome [11].

Ghana, particularly, the Greater Accra region is endemic for malaria. In 2017, Ghana National Malaria Control Program report indicated that almost 48% of all out-patient department attendants were attributable to malaria [12]. Whereas in the Greater Accra region, prevalence of malaria has been reported to be 15.1% [13]. In the region, malaria mostly affect children less than 15 years, males, rural and peri-urban dwellers as well as people with either no or only primary education. Additionally, unemployed and people engaging in petty trading with lower incomes are disproportionately affected [14]. Over 95% of malaria cases in the region is attributable to the *P. falciparum* spp, while the rest are caused by either *P. malariae* or *P. ovale* [15].

Both malaria and sickle cell disease are blood-associated diseases that affect the hematological cell lines [16, 17] but with varied clinical presentations. Whereas fever, cephalgia, fatigue, malaise, and musculoskeletal pain are commonly seen in uncomplicated malaria [18], the clinical features of homozygous sickle cell disease varies widely between patients [19].

Malaria is associated with oxidative stress where there is generation of large amounts of reactive oxidative and nitrogen stress which cause an imbalance between the formation of oxidizing species and the activity of antioxidants [20]. In previous studies in Ghana and elsewhere, malondialdehyde was found to be markedly elevated in malaria [21, 22]. Similarly, sickle cell disease is also associated with oxidative stress [23]. However, very few studies have reported oxidative stress in sickle cell patients with malaria, but none, to the best of our knowledge has been reported in Ghana. Therefore, this study assessed the degree of oxidative stress in SCD patients with malaria and the associated hematological and disease presentation profile in Ghanaian patients. In this study, the isoprostane, 8-iso-prostaglandin F2α (8-iso-PGF2α), was used to assess the relative levels of oxidative stress. Measurement of 8-iso-PGF2α is a reliable tool for assessing enhanced rates of lipid peroxidation in disease states [24]. The isoprostanes have been widely used as a reliable biomarker of oxidative stress [25]. Among the three major classes of isoprostanes, the F2 class has been recognized as the most suitable biomarker since D2 and the E2 are less stable [26]. F2-isoprostanes have been used as a biomarker for oxidative stress and their levels have been measured in a wide range of biological samples such as urine, plasma, and exhaled breath condensate [27].

## Materials and methods

### Study design, study site, study period, and ethical considerations

This prospective cohort study took place at the Ga North Municipal Hospital, Ofankor, in the Greater Accra region of Ghana between August 2018 and July 2019. The Ga North Municipal

Hospital is a public referral health facility that sees an average of over 200 patients a day. The hospital is a referral hospital for several smaller public and private health centers in the municipality. The hospital operates an out-patient department, in-patients department, antenatal services as well as infectious and non-infectious diseases clinic. Of the average number of patients seen daily, the average malaria cases recorded per day is about seven while the non-communicable section of the hospital sees about four sickle cell patients in a day. Individuals in each study cohorts were randomly selected during the study period, until the pre-determined sample size was achieved. Ethical approval for the study was granted by Ghana Health Service Ethics Review Committee (Approval No: GHS-ERC002/03/18). Study participants over 18 years of age provided written informed consent whereas parental assent was obtained from participants less than 18 years of age. Declaration of Helsinki was strictly followed in this study, regarding ethical issues and recruitment of study participants in accordance with applicable local laws.

## Sample collection strategy

On each day, a maximum of three individuals with microscopically detectable malaria were systematically selected for the study. The clinical records of the consented patients were reviewed for clinical and laboratory findings associated with the current morbidity. Urine sample (approx. 20 mL) was collected from each consented patient for urinalysis. Plasma was separated from whole blood, stored below -30°C, until biochemical and 8-iso-prostaglandin F2α levels analyses were performed.

## Study subjects, sample size, and sample processing

Subjects investigated in this study were individuals with malaria with homozygous hemoglobin A (malaria cohort), individuals with both malaria and sickle cell disease (HbSS) (malaria-SCD cohort) and individuals with sickle cell without malaria (SCT cohort). Sample size was determined based on single population formula using confidence interval of 95% and an estimated previous prevalence of sickle cell disease (SCD) in malaria of 50% (prevalence unknown). The sample size was calculated to be 385. Based on the interquartile range of individuals with both malaria and SCD, an equal number of normal controls (individuals without malaria and SCD) and SCD without malaria were selected for the study from the study samples that satisfied the inclusion criteria. Hematological profile was done on the same day of sample collection while sickle cell phenotyping was done later.

## Inclusion and exclusion criteria

Patients included in the study were microscopy diagnosed malaria patients with either self or parental consent. To be able to obtain self-reported clinical history, only patients older than 10 years were included in this study. Persons who received any parenteral fluid were excluded. Additionally, all SCD patients that visited the hospital on account of sickle cell crisis as well as those infected with hepatitis B virus, hepatitis C virus, syphilis, and HIV were excluded. Samples with sickle cell traits were also excluded because very few of them were encountered in this study.

## Selection of comparative groups

Individuals with sickle cell disease without malaria (n = 45), and blood samples of patients with malaria but without SCD were carefully selected for inclusion based on the age range

obtained for patients with sickle cell disease with malaria. Patients with SCD (HbSS) in steady state were recruited during sickle cell clinic days.

## Laboratory analysis

**Malaria parasite detection and quantification.** Thick and thin blood films were done on the same glass slide for each specimen, in triplicate. The dried thin film was fixed in absolute methanol, briefly, air dried, and stained with 10% Giemsa. The stained smears were examined for the presence of *Plasmodium* parasites. The parasites were identified to the species level and quantified per μL of blood according to the following WHO guidelines. Briefly, parasites were quantified per 200 WBCs counted using the patients' total WBCS per μL of blood. Each slide was double checked by a blinded certified malaria microscopist and in cases of discordant results, in terms of speciation and parasite count, a third opinion closer to any of the two was final.

**Infectious marker screening.** Each blood sample was screened for other infectious markers, namely, hepatitis B virus, hepatitis C virus, syphilis, and HIV. The infectious marker screening was done with rapid immunochromatographic test kits. HIV and syphilis were screened with First Response® test kit (Premier Medical Corporation Ltd, India) while the hepatitis B and C were screened with Wondfo Rapid Diagnostic Test (Guangzhou Wondfo Biotech Co. Ltd, China).

**Hematological profiling.** Hematological profiling was done using Urit 5200 (Urit Medical Electronic Co. Ltd, China) fully automated hematology analyzer. The 5-part differential analyzer works on the principle of laser beam multidimensional cell classification, flow cytometry for white cell differentiation, and white and red blood cell estimation. Platelets were counted by optical and electrical impedance principles and hemoglobin concentration was measured by cyanide free colorimetric method. All other parameters were calculated.

**Determination of kidney function parameters and plasma bilirubin fractions.** Plasma bilirubin and creatinine were assayed with PKL-125 (Italy) biochemistry analyser using default settings at 546 nm and 505 nm, respectively. Urine creatinine was assayed using the same analyzer but the sample was diluted 1 in 50 (1 part of urine sample to 49 parts of distilled water). Measured concentration was adjusted using the dilution factor to obtain the final urinary creatinine concentration.

**Sickle cell screening and phenotyping.** Sickle cell screening was done by HemoTypeSC immunochromatographic test kit (Silver Lake Research Corporation Ltd, USA). The manufacturer's instruction was strictly followed. All study participants with malaria were screened for sickle cell hemoglobin. Sickle cell haplotypes were identified as recently published [28].

**Sandwich-ELISA for 8-iso-prostaglandin F2alpha levels.** Reagents and consumables for 8-iso-prostaglandin F2α was obtained from SunLong Biotech (Hangzhou, China, Catalogue Number: SL0035Hu). Measurement of 8-iso-prostaglandin F2α was done according to the manufacturer's protocol, with slight modification. Colorimetric measurement of 8-iso-prostaglandin F2α was based on the principle that the micro-ELISA strip plates were precoated with an antibody specific to human 8-iso-prostaglandin F2α. Fifty (50) microliters of pre-diluted samples were used in the procedure. All other volumes of reagents were used as directed by the manufacturer. The optical densities were measured by Mindray MR-96A ELISA microplate reader (Shenzhen, China) at a wavelength of 450 nm with plate correction at 630 nm. The concentrations of 8-iso-prostaglandin F2α were obtained automatically by preprogrammed assay curve.

## Statistical analysis

Differences in the mean of continuous parametric data were determined by ANOVA and multiple comparison was done with Tukey modelling test. Chi square was used to determine the

association of clinical history and urinalysis findings of the patients with malaria or malaria with SCD whereas differences in parametric variables were determined by t-test. P-value of < 0.05 was considered significant. Correlation between variables was determined by Pearson R. Statistical analyses were done with Stata 15.0 statistical software.

## Results

### Characteristics of the study participants analyzed in this study

A total of 411 study participants with malaria satisfied the inclusion criteria for this study. Of this number, 45 participants with sickle cell disease (SCD) and not infected with any of the other infectious markers screened were included in the study. From the rest of the malaria cases with SCD, an equal number of patients (n = 45) were selected to match the malaria-SCD cohort. Additionally, 45 patients (53.3% females) with SCD without malaria were recruited as a comparative cohort. There were slight differences between the median ages (in years) of the three cohorts (p = 0.418), with the median age of the malaria cohort marginally higher. The mean body temperature of the malaria-SCD cohort was significantly higher than the other two cohorts (38.0±0.87˚C, p = 0.019). Furthermore, the lower and upper ranges of malaria parasitemia were higher in the malaria cohort than malaria-SCD cohort (Table 1).

### Plasma levels of 8-iso-prostaglandin F2α in relation to age and gender

The 8-iso-prostaglandin F2α (8-iso-PGF2α) levels in the three cohorts are presented in Table 2. The 8-iso-PGF2α levels were significantly different among the age ranges, gender, and various ranges of parasite density across the three cohorts. Additionally, it was observed that 8-iso-PGF2α levels were numerically higher in the 10–20-year age range than the 20–29-year group, irrespective of the cohort. Furthermore, in the three cohorts, 8-iso-PGF2α levels were higher in males than in females, whereas parasite densities increased with 8-iso-PGF2α levels in the malaria parasite-infected cohorts. However, the percentage increases in the 8-iso-PGF2α levels were higher in malaria-SCD cohort than the malaria cohort (49.8% vs. 29%) even though the geometric mean of the parasite density was higher in the malaria cohort than the malaria-SCD cohort. In spite of the above observation, the 8-iso-PGF2α levels in malaria-SCD cohort were significantly higher in all study variables compared to the other groups.

**Table 1. Demographic, temperature, and parasitemia of the patients.**

| Variables | Malaria-HbAA (n = 45) | Malaria–SCD (n = 45) | SCD (n = 45) | p-value |
|---|---|---|---|---|
| IQR of age (years) | 14–27 | 13–24 | 15–28 | |
| Median age (years) | 19 | 16 | 17 | 0.047 |
| *Gender* | | | | 0.418 |
| Males, n (%) | 27 (60.0) | 21 (46.7) | 21 (46.7) | |
| Females, n (%) | 18 (40.0) | 24 (53.3) | 24 (53.3) | |
| IQR of body temperature (˚C) | 36.9–38.9 | 37–40.9 | | |
| Body temp range (mean±SD) | 37.3±1.15 | 38.0±0.87 | 36.7±0.22 | 0.019 |
| Parasite density range (/μL) | 10,213–320,586 | 2,492–142,452 | | |

Malaria-HbAA-malaria in individuals with HbAA haplotype; Malaria-SCD-malaria in individuals with HbSS haplotype; SCD-sickle cell disease.

**Table 2. Analysis of 8-iso-prostaglandin F2α levels in malaria and SCD.**

| Variables | Mean plasma levels of 8-iso-prostaglandin F2α (pg/mL) | | | |
| --- | --- | --- | --- | --- |
| | Malaria-HbAA | Malaria–SCD | SCD | p-value |
| *Age (years)* | | | | |
| **10–19** | 173.9±17.1 | 233.1±21.7 | 113.3±10.5 | < 0.001 |
| **20–29** | 119.3±23.7 | 211.3±19.4 | 99.1±6.6 | < 0.001 |
| *Gender* | | | | |
| **Male** | 182.1±20.1 | 281.6±29.0 | 108.5±8.7 | < 0.001 |
| **Female** | 111.1±19.6 | 162.8±25.5 | 103.9±12.3 | < 0.001 |
| *Parasite density sub-range* | | | | |
| **10,000–50,000** | 127.5±13.1 (n = 14) | 149.9±17.4 (n = 19) | Not applicable | < 0.001 |
| **50,001–100,000** | 130.1±11.5 (n = 14) | 217.0±22.1 (n = 15) | Not applicable | < 0.001 |
| **> 100,000** | 179.6±20.3 (n = 17) | 298.7±19.6 (n = 12) | Not applicable | < 0.001 |
| **Overall mean** | 146.6±22.31 | 222.2±30.05 | 106.2±19.4 | < 0.001 |

## Clinical presentation and association of urinalysis findings with disease conditions

More malaria-SCD cohorts than malaria cohorts reported chills (84.4% vs. 71.1%, p = 0.128) and headache (42.2% vs. 35.6%, p = 0.517) but the difference was not statistically significant. However, fever (p = 0.0491), anorexia (p = 0.043), joint pain (p < 0.001), lethargy (p = < 0.001) and vomiting (p = 0.029) were significantly associated with malaria-SCD (Table 3).

## Urine biochemistry of the study cohorts

The urine of malaria-SCD cohorts was slightly acidic compared to the malaria cohort. Additionally, the urine of 80% of malaria-SCD cohort was hypersthenuric. Whereas ketonuria (p = 0.250) was not associated with any of the cohorts, gross hematuria (< 0.001), microhematuria (< 0.001), proteinuria (< 0.001) and bilirubinuria (< 0.001) were highly associated with malaria-SCD (Table 4).

**Table 3. Clinical history of the patients.**

| Clinical manifestations | Response | Malaria-HbAA | Malaria-SCD | $x^2$ (p-value) |
| --- | --- | --- | --- | --- |
| Chills | Present | 32 (71.1) | 38 (84.4) | 0.128 |
| | Absent | 13 (28.9) | 7 (15.5) | |
| Fever | Present | 39 (86.7) | 40 (88.9) | 0.0491 |
| | Absent | 6 (13.3) | 5 (11.1) | |
| Anorexia | Present | 31 (68.9) | 39 (86.7) | 0.043 |
| | Absent | 14 (31.1) | 6 (13.3) | |
| Headache | Present | 16 (35.6) | 19 (42.2) | 0.517 |
| | Absent | 29 (64.4) | 26 (57.8) | |
| Joint pains | Present | 11 (24.4) | 39 (86.7) | < 0.001 |
| | Absent | 34 (75.6) | 6 (13.3) | |
| Lethargy | Present | 17 (37.8) | 36 (80.0) | < 0.001 |
| | Absent | 28 (62.2) | 9 (20.0) | |
| Vomiting | Present | 23 (51.1) | 37 (82.2) | 0.029 |
| | Absent | 22 (48.9) | 12 (17.8) | |

**Table 4. Urinalysis findings of the patients.**

| Urine parameters | Results | Malaria-HbAA | Malaria-SCD | $x^2$ (p-value) |
|---|---|---|---|---|
| Gross hematuria | Present | 2 (4.4) | 31 (68.9) | < 0.001 |
| | Absent | 43 (95.6) | 14 (31.1) | |
| Microhematuria | Present | 9 (20.0) | 38 (84.4) | < 0.001 |
| | Absent | 36 (80.0) | 7 (15.5) | |
| Proteinuria | Present | 14 (31.1) | 35 (77.8) | < 0.001 |
| | Absent | 31 (68.9) | 10 (22.2) | |
| Bilirubinuria | Present | 7 (15.5) | 34 (75.6) | < 0.001 |
| | Absent | 38 (84.4) | 11 (24.4) | |
| Ketonuria | Present | 11 (24.4) | 16 (35.6) | 0.250 |
| | Absent | 34 (75.6) | 29 (64.4) | |
| Glucosuria | Present | 0 (0.0) | 0 (0.0) | [a] |
| | Absent | 45 (100) | 45 (100) | |
| Specific gravity | Normal | 19 (42.2) | 9 (20) | |
| | Hypersthenuria[b] | 26 (57.8) | 36 (80.0) | 0.023 |
| Mean pH | | 6.8 | 6.5 | 0.058 |

[a] Chi statistic indeterminate

[b] Hypersthenuria is urine specific gravity > 1.025.

## Association of hematological parameters with malaria, SCD, and malaria-SCD cohort

Leukocytosis, eosinophilia, monocytosis, low red blood cell count, low hemoglobin level, low hematocrit, low MCV, low MCH and low MCHC were associated with malaria-SCD cohort whereas thrombocytopenia and low plateletcrit were associated with malaria cohort. SCD cohort was only associated with relative lymphocytosis. Counts of neutrophils, basophils, mean platelet volume, and platelet large cell ratio did not differ across the three cohorts (Table 5).

## Predictive cellular inflammatory biomarkers in malaria-SCD comorbidity

Leukocyte ratios differed significantly across the three cohorts. However, low neutrophil-eosinophil ratio (NER), low lymphocyte-eosinophil ratio (LER), high monocyte-basophil ratio (MBR) and very high eosinophil-basophil ratio (EBR) were associated with malaria-SCD cohort (Table 6). Among the four leukocyte ratios significantly associated with malaria-SCD, none of the raw computed ratios of neutrophils/eosinophils for malaria and SCD cohorts was lower than 18 and none of the eosinophils/basophil exceeded 10. Therefore, using NER < 18 and EBR > 10 cut off points, their respective sensitivities were 80% (36/45) and 87% (39/45).

## Kidney function and hemoglobin metabolism in malaria and SCD disease

Significant increases in creatinine and bilirubin levels were observed in the sickle cell cohorts, however, higher levels were obtained in the comorbid state. Additionally, urea level was significantly lower in malaria cohorts while in the SCD and comorbid states, the differences were not significant, although a lower mean urea level was observed in the comorbid state. Furthermore, spot urinary creatinine was significantly lower in the comorbid states with malaria cohorts recording higher levels than the SCD patients (Table 7).

**Table 5. Hematological parameters associated with malaria, SCD and malaria-SCD comorbidity.**

| Hematological parameters | Malaria-HbAA (mean±SD) | Malaria-SCD (mean±SD) | SCD (mean±SD) | p-value |
|---|---|---|---|---|
| WBC (x10$^9$/L) | 6.68±2.42 | 18.32±2.77* | 9.91±2.01 | <0.001 |
| Neutrophils (%) | 62.1 ±10.1 | 50.44±8.65 | 43.99±5.33 | 0.081 |
| Lymphocytes (%) | 28.53±8.22 | 36.23±8.44 | 41.0±3.09* | <0.001 |
| Eosinophils (%) | 2.19±1.79 | 4.77±0.99* | 1.21±1.01 | <0.001 |
| Monocytes (%) | 5.92±3.30 | 7.32±1.58* | 3.56±2.12 | 0.021 |
| Basophils (%) | 0.45± 0.24 | 0.32±0.17 | 0.51±0.11 | 0.073 |
| RBC (x10$^{12}$/L) | 4.22±0.78 | 2.87±1.04* | 3.76±0.69 | 0.018 |
| Hemoglobin (g/dL) | 10.83 ±2.11 | 7.19 ±1.06* | 10.11±2.00 | <0.001 |
| Hematocrit (%) | 31.84 ±6.07 | 21.34±2.79* | 29.27±4.31 | <0.001 |
| Mean Cell Volume | 76.07 ±5.53 | 69.45±6.21* | 71.33 ±7.62 | 0.015 |
| MCH (pg) | 25.89 ±3.78 | 22.19±3.11* | 24.53 ±4.09 | 0.0041 |
| MCHC (g/dl) | 34.07 ±2.35 | 27.71±2.85* | 31.02±3.08 | <0.001 |
| RDW_CV (%) | 14.29 ±1.78 | 14.48 ±1.68 | 15.98±2.86 | 0.485 |
| Platelets (x10$^9$/L) | 138.71± 27* | 190.0± 15.31 | 245.0±28.33 | <0.001 |
| MPV (fL) | 9.91 ±1.40 | 10.78 ±1.28 | 12.06±2.30 | 0.091 |
| Plateletcrit (%) | 0.18±0.08* | 0.22±0.04 | 0.27±0.09 | <0.001 |
| P_LCR | 30.21±6.39 | 25.11±4.06 | 29.21±4.31 | 0.052 |

MCHC = Mean cell hemoglobin concentration, MCH = Mean cell hemoglobin, MCV = Mean cell volume, RDW_CV = Red cell distribution width coefficient of variation, RDW_SD = Red cell distribution width standard deviation, L = Litre, fL = Fentolitre, pg = pictogram, Plt = Platelets, PDW = Platelet distribution width, P_LCR = Platelet large cell ratio.

*Significantly different variables.

## Discussion

This study reports the elevation of 8-iso-prostaglandin F2α (8-iso-PGF2α), an oxidative stress biomarker in malaria and sickle cell disease (SCD) comorbidity and associated laboratory and clinical factors. Significant increase in 8-iso-PGF2α oxidative stress marker corresponded to a significant increase in body temperature as well as feverishness. This association has previously

**Table 6. Association of leukocyte ratio with malaria-SCD comorbidity.**

| Leukocyte ratios | Malaria-HbAA | Malaria-SCD | SCD | p-value |
|---|---|---|---|---|
| NLR | 2.18±1.23 | 1.39±1.02 | 1.07±1.72* | < 0.001 |
| NER | 28.36±5.64 | 10.57±8.74* | 36.36±5.28 | < 0.001 |
| NMR | 10.49±3.06 | 6.89±5.47* | 12.36±2.51 | < 0.001 |
| LER | 13.03±4.59 | 7.60±3.53* | 33.88±3.06 | < 0.001 |
| LMR | 4.82±2.49 | 4.95±1.34 | 11.52±1.46* | < 0.001 |
| LBR | 63.40±11.34* | 93.22±14.47 | 80.39±28.09 | < 0.001 |
| EMR | 0.37±0.05 | 0.45±0.04* | 0.34±0.11 | < 0.001 |
| EBR | 4.87±2.33 | 14.91±4.14* | 2.37±0.18 | < 0.001 |
| MBR | 13.16±4.12 | 22.88±2.57* | 6.98±1.27 | < 0.001 |

NLR-neutrophils-lymphocytes ratio; NER-neutrophils-eosinophils ratio; NMR-neutrophils-monocytes ratio; NBR-neutrophils-basophils ratio; LER-lymphocytes-eosinophils ratio, LMR-lymphocytes-monocytes ratio, LBR-lymphocytes-basophils ratio; EMR-eosinophils-monocytes ratio; EBR-eosinophils-basophils ratio, MBR-monocytes-basophils ratio.

*Significantly different variable.

**Table 7. Urea, creatinine and bilirubin levels in the study participants.**

| Markers | Malaria-HbAA | Malaria-SCD | SCD | p-value |
|---|---|---|---|---|
| Plasma | | | | |
| Urea (mmol/L) | 5.3±2.1* | 7.9±3.9 | 8.3±2.7 | < 0.001 |
| Creatinine (μmol/L) | 71.7±5.1 | 203.0±8.3* | 106.5±9.2* | < 0.001 |
| Bilirubin (total) (μmol/L) | 17.8±2.2 | 161.3±7.6* | 35.3±6.1* | < 0.001 |
| Bilirubin (unconjugated) (μmol/L) | 12.7±3.1 | 145.1±10.5* | 28.1±9.5* | < 0.001 |
| Urine (spot sample) | | | | |
| Creatinine (μmol/L) | 9033±213 | 3212±1671* | 4,111±701* | < 0.001 |

been established [29, 30] together with elevated pro-inflammatory cytokines in such cases [31]. These two; elevated proinflammatory cytokines and oxidative stress, have been implicated in pathophysiological events in non-communicable diseases [32–34] and communicable diseases [35–37] as well.

In spite of the fact that body temperature was significantly elevated in malaria-SCD cohort than the malaria cohort, the range of malaria parasitemia was higher in malaria cohorts compared to malaria-SCD cohorts. Even though SCD patients are very vulnerable to malaria [38], levels of parasitemia are always low [39] compared to individuals without hemoglobin S. This observation could be due to splenic removal of abnormally-parasitized red cells from the body [40, 41]. Another reason could be reduced parasite replication in HbS erythrocytes [42]. Additionally, it is suggested that reduced hemoglobin solubility under low oxygen tension and increased sickling of parasitized cells enhances the reduction of malaria parasite density in the reticuloendothelial system [43].

It was further observed that malaria exacerbated oxidative stress in SCD, irrespective of age, gender, or parasite density. This observation could be as a result of the cumulative effect of oxidative stress contributed individually by both the *Plasmodium falciparum* acute infection and sickle cell status. It is obvious to note that in malaria-SCD, excessive production of free radicals overwhelms the neutralization capacity of antioxidants. This results in enhanced lipid peroxidation which plays an important role in disease pathogenesis and presentation [44]. This harmful processes cause massive damages to biomolecules such proteins, lipids, and nucleic acids [45]. Hence, the significant deviations of hematological parameters in the comorbidity state from the single morbidity states could be attributed to the significant elevation of 8-iso-PGF2α, the oxidative stress biomarker. Total white blood cells, eosinophils, and monocytes subpopulations were significantly elevated in the comorbidity state. In a steady state, the leukocytes count of SCD patients in Ghana was reported to be 12.1 x10$^9$/L whereas those in vaso-occlusive crisis (VOC) was reported as 16.2 x10$^9$/L [17]. In this study, the leucocyte count in malaria-SCD was 18.3 x10$^9$/L, a value slightly higher than the count previously reported for SCD in VOC. This confirms an earlier publication that leukocytes count are expected to increase in SCD with any form of complications [46]. Additionally, most of the erythrocyte indices were significantly lower in malaria-SCD, which is consistent with severe anemia. Even though the numerical levels of these parameters were low in malaria and SCD cohorts, malaria worsened the levels in SCD. Although malaria parasitemia is consistent with thrombocytopenia [47], malaria parasitemia did not worsen the levels in SCD. Four leukocyte ratios, neutrophils-eosinophils ratio (NER), lymphocytes-eosinophils ratio (LER), eosinophils-basophils ratio (EBR) and monocytes-basophils ratio (MBR) were found to be associated with malaria-SCD, however, NER < 18 and EBR > 10 were respectively sensitive (80% and 87%). Therefore, EBR cut-off value greater than 10 could be used to predict malaria in SCD, provided the sickle

cell patient is in stable condition without vaso-occlusive crises. These ratios could be diagnostic because mean proportion of eosinophils were significantly higher in comorbid state whereas in mean basophils proportion were lower in comorbid state, even though it did not reach significant level. Leukocyte ratios have been used to predict several diseases in clinical practice [48, 49]. But this is the first time, to the best of our knowledge, leukocyte ratios are being studied in sickle cell disease and malaria comorbidity. Therefore, this concept needs to be proved in future studies in other endemic areas. Aside the diagnostic potential of EBR in malaria-SCD comorbidity, the EBR ratio has been reported to be increased in inflammatory response [50]. Therefore, elevated EBR ratios and 8-iso-PGF2α biomarker will eventually elevate the inflammatory response in malaria-SCD comorbidity. A situation that could trigger VOC.

Although an appreciably high numbers of the malaria cohort presented with chills, fever, anorexia, headache, joint pains, lethargy, and vomiting, 80% or more malaria-SCD cohort presented with these symptoms except headache. This indicates that malaria may impact severely on sickle cell disease status, with simultaneous presentation of several symptoms, such discomfort cannot be underestimated.

Analysis of the urine samples in the two cohorts revealed interesting findings. Gross hematuria, microhematuria, proteinuria, and bilirubinuria were detected in high frequencies (68.9–84.4%) in the malaria-SCD cohort. Additionally, hypersthenuria and low pH were also observed. Significantly, these urine parameters associated with malaria-SCD. Based on the foregoing, malaria parasites could be said to worsen kidney function in sickle cell patients. Even though plasma creatinine and bilirubin, and urinary creatinine levels were respectively, higher and lower than physiological levels in SCD cohorts, significantly elevated levels of plasma creatinine and bilirubin were observed in malaria-SCD cohort. Correspondingly, lower urinary creatinine in malaria-SCD cohort confirmed impaired renal function in malaria-SCD cohort. Unfortunately, we could not analyze for creatinine clearance due to difficulty in obtaining 24-hour urine samples of the study participants. That notwithstanding, elevations in mean plasma creatinine and reduction in spot urine creatinine could confirm reduced kidney function as has been previously reported in several studies [51–53].

Critical analysis of SCD patients with malaria suggested the presence of intravascular hemolysis. This was evidenced by significantly low red blood cells with its accompanying low hemoglobin levels. Additionally, unconjugated bilirubin level was significantly elevated together with gross and microhematuria as well as bilirubinuria. It is obvious that low hemoglobin observed was as a result of red cell break down which was confirmed by low red blood cell count. Profuse hemolysis observed in comorbid state was not surprising since 8-iso-PGF2α oxidative stress biomarker, which was elevated in SCD-malaria, has been associated with hemolysis [54]. Enhanced metabolism of the hemoglobin yielded more bilirubin which overwhelmed the conjugation ability of the liver. With this observation, unconjugated jaundice could be common in sickle cell patients with malaria. In the face of elevated 8-iso-PGF2α, majority of the comorbid patients presented with chills, fever, anorexia, joint paints, lethargy and vomiting. It could therefore be suggested that malaria could trigger acute hemolytic crisis in sickle cell patients.

## Conclusion

Malaria in sickle cell disease is a medical emergency. It is therefore, recommended that all sickle cell patients experiencing hemolytic crisis should be tested for malaria. This will ensure prompt management to prevent imminent VOC. Furthermore, thorough and adequate liver and renal assessments in sickle cell patients with malaria are essential to halt and reverse further organ damage.

## Limitations

We could not assess creatinine clearance in the patients due to difficulties in 24-hour urine output of the patients. Additionally, electrolyte levels were not assessed because we did not want to collect additional blood samples after about 5 mL had been collected for routine laboratory analysis.

## Supporting information

**S1 Data.**
(XLSX)

## Acknowledgments

We are grateful to Alex Nyarko, Derrick Kwarteng, Samuel Ohene Ofori, and Irene Serwaa Ofosu for the various roles they played as research assistants to collect primary data for this study.

## Author Contributions

**Conceptualization:** Enoch Aninagyei, Desmond Omane Acheampong.

**Data curation:** Henrietta Kwansa-Bentum, George Ghartey-Kwansah, Alex Boye.

**Formal analysis:** Henrietta Kwansa-Bentum, Adjoa Agyemang Boakye, George Ghartey-Kwansah.

**Investigation:** Enoch Aninagyei, Clement Okraku Tettey, Adjoa Agyemang Boakye, George Ghartey-Kwansah, Alex Boye.

**Methodology:** Enoch Aninagyei, Clement Okraku Tettey, Adjoa Agyemang Boakye, George Ghartey-Kwansah, Alex Boye.

**Project administration:** Enoch Aninagyei, Alex Boye, Desmond Omane Acheampong.

**Supervision:** Desmond Omane Acheampong.

**Validation:** Henrietta Kwansa-Bentum, Adjoa Agyemang Boakye.

**Visualization:** Adjoa Agyemang Boakye.

**Writing – original draft:** Enoch Aninagyei, Clement Okraku Tettey, Henrietta Kwansa-Bentum, Adjoa Agyemang Boakye, George Ghartey-Kwansah.

**Writing – review & editing:** Enoch Aninagyei, Clement Okraku Tettey, Henrietta Kwansa-Bentum, Adjoa Agyemang Boakye, George Ghartey-Kwansah, Alex Boye, Desmond Omane Acheampong.

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
