## [Decision Letter · Decision Letter 0]

21 Feb 2022

PONE-D-21-30010Oxidative stress and associated clinical manifestations in malaria and sickle cell (HbSS) comorbidityPLOS ONE

Dear Dr. Enoch Aninagyei,

Thank you for submitting your manuscript to PLOS ONE. After careful consideration, we feel that it has merit but does not fully meet PLOS ONE’s publication criteria as it currently stands. Therefore, we invite you to submit a revised version of the manuscript that addresses the points raised during the review process.

Please consider enriching the instruction section by including pathophysiology of SCD and the hemolytic aspect. Also improve on the epidemiology of malaria.In the methodology section could you please include the functioning of the hospital and the number of patients per diseases conditions.Also define the study design. On the discussion, please elaborate more on leukocyte ratio and disease condition, justify the markers for SCD and discuss more on hemolysis.

We look forward to receiving your revised manuscript.

Kind regards,

Gabriel Agbor

Academic Editor

PLOS ONE

https://journals.plos.org/plosone/s/fileid=ba62/PLOSOne_formatting_sample_title_authors_affiliations.pdf".

“No external funding was obtained for this study. The study was funded from authors' resources.”

Reviewers' comments:

Reviewer's Responses to Questions

**Comments to the Author**

1. Is the manuscript technically sound, and do the data support the conclusions?

Reviewer #1: Yes

Reviewer #2: Partly

2. Has the statistical analysis been performed appropriately and rigorously? 

Reviewer #1: Yes

Reviewer #2: Yes

3. Have the authors made all data underlying the findings in their manuscript fully available?

Reviewer #1: Yes

Reviewer #2: Yes

4. Is the manuscript presented in an intelligible fashion and written in standard English?

Reviewer #1: Yes

Reviewer #2: No

5. Review Comments to the Author

Reviewer #1: I would like to congratulate the authors for taking this personal initiative to understand the clinical manifestations in these disease conditions. However, I have enumerated a few points for the authors to address and upgrade the manuscript.

1)Was the study both a retrospective and prospective Cohort study given that data was collected from people already suffering from SCD coming to the clinic for visits and people diagnosed with malaria and SCD.

2)Under study design you wrote its a cross-sectional study, could you please state the appropriate study design

3)Given that Malaria infectivity varies seasonally, were the samples collected at same peak seasonal period or randomly? This is to minimize variations in samples parameters

4)In the discussion section, please elaborate more on the leukocyte ratio effect on the disease condition results

5)Was informed consent obtained before engaging the patients in the study. If yes please clearly state that and precise the age groups from which consent was obtained from

6) You stated in the manuscript that `the oxidative stress status of sickle cell patients with malaria has not been studied’. Provide more clarity on this e.g see publication below

Atiku SM, Louise N, Kasozi DM. Severe oxidative stress in sickle cell disease patients with uncomplicated Plasmodium falciparum malaria in Kampala, Uganda. BMC Infect Dis. 2019;19(1):600. Published 2019 Jul 9. doi:10.1186/s12879-019-4221-y

Reviewer #2: The present study proposes to carry out a study that investigates the impact of condition of having concomitant sickle cell disease and malaria, comparing with individuals with malaria and with sickle cell disease, describing biomarkers associated with renal, hematological, and oxidative stress changes and clinical manifestation. The subject is very important, mainly in the area that both disease occur frequently.

The authors should include in their Introduction, more about SCD pathophysiology, and about the hemolytic aspect of the disease, that is hereditary; also, and about epidemiology of malaria in the region. What kind of Plasmodium is common in the region? The methodology is well described but will be important place about how the hospital work, and how many patients of each disease they receive. The authors include Bilirubin as a renal marker, it is necessary to explain it, because there is a mistake about this data. About the ethics aspect, as they included patients over 10 years old, it will be important to place the parental consent and that the Declaration of Helsinki was followed. The authors need to place clearly that comorbidity is related to the presence of malaria and SCD.

In the results, authors should include more specific data, and confirm data about age in the table 1 and in the text. In table 2, will be important to include how many patients they found in each group of parasite density sub-range. The results are well presented, but need to correct some mistake, such as, the authors referrer leukocytes as level and not count.

The discussion needs to be rewritten the discussion, and please they need to include more about the hemolysis marker investigated at the study, there are several markers, but the authors need to justify their choice. It is important because, there is a paper that reports about the influence of hemolysis in the 8-iso-prostaglandin F2α levels, that need to be incorporated at the discussion section (Ulrike Dreiβigacker et al. Clinical Biochemistry 43 (2010) 159–167). When the authors talk about the “EBR cut-off value greater than 10 could be used to predict malaria in SCD”, there is a several important aspect that need to be incorporate such as the SCD crisis. In addition, they need to include the kidney hyperfiltration that occur in SCD, mainly in the HbSS genotype.

6. PLOS authors have the option to publish the peer review history of their article (what does this mean?). If published, this will include your full peer review and any attached files.

Reviewer #1: No

Reviewer #2: No

---

## [Author Response · Author response to Decision Letter 0]

15 Mar 2022

Editorial review

Comment: Please consider enriching the instruction section by including pathophysiology of SCD and the hemolytic aspect. 

Response: The introduction section has been improved by adding the suggested information. The following information has been added ‘HbS result from the substitution of hydrophilic glutamic acid by hydrophobic valine at the sixth position in the β-globin chain (5). Globally, 3.2 million people live with HbSS or HbSC. About 176,000 people die of HbS disease related complications every year (6). Anaemia is common in sickle patients (7) together with vaso-occlusion which frequently leads to ischemia. This cascade of events is the predominant pathophysiology responsible for acute painful vaso-occlusive crisis (8). Increased plasma viscosity occurs as a result of chronic hemolysis and reduced sickle red cell deformability due to HbS polymerization (9). These effects could be prevented or reversed by therapies that prevents HbS polymerization by allosterically modifying HbS oxygen affinity, preventing erythrocyte dehydration. Hydroxyurea, metformin and sodium butyrate are common examples (10)’ 

Comment: Also improve on the epidemiology of malaria.

Response: Epidemiology of malaria has been added. Ghana, particularly, the Greater Accra region is endemic for malaria. in 2017 Ghana National Malaria Control Program report indicated that almost 48% of all Out-patient department attendants were attributable to malaria (5). Whereas in the Greater Accra region, prevalence of malaria has been reported to be 15.1% (6). In the region, malaria mostly affect children less than 15 years, males, rural and peri-urban dwellers as well as people with either no or only primary education. Additionally, unemployed and people engaging in petty trading with lower incomes are disproportionately affected (7) (page 1). 

Comment: In the methodology section could you please include the functioning of the hospital and the number of patients per diseases conditions.

Response: The manuscript has been revised accordingly to include these statements ‘The Ga North Municipal Hospital is a public referral health facility that sees an average of over 200 patients a day. The hospital is a referral hospital for several smaller public and private health centres. The hospital operates an out-patient department, in-patients department, antenatal services as well as infectious and non-infectious diseases clinic. Of the average number of patients seen daily, malaria cases recorded per day is about 7 while the non-communicable section of the hospital sees about four sickle cell patients a day. Individuals in each study cohorts were randomly selected during the study period, until the pre-determined sample size was achieved’. 

Comment: Also define the study design. 

Response: The study design has been changed to prospective cohort study (page 4)

Comment: On the discussion, please elaborate more on leukocyte ratio and disease condition, justify the markers for SCD and discuss more on hemolysis.

Response: The discussion section has been improved with more information on leukocyte ratios. This information was added to the revised manuscript ‘These ratios could be diagnostic because mean proportion of eosinophils were significantly higher in comorbid state whereas in mean basophils proportion were lower in comorbid state, even though it did not reach significant level. Leukocyte ratios have been used to predict several diseases in clinical practice (48,49). But this is the first time, to the best of our knowledge, leukocyte ratios are being studied in sickle cell disease and malaria comorbidity’

Additionally, the discussion section has been revised to include more on hemolysis. The following were added ‘Critical analysis of SCD patients with malaria suggested the presence of intravascular hemolysis. This was evidenced by significantly low red blood cells with its accompanying low hemoglobin levels. Additionally, unconjugated bilirubin level was significantly elevated together with gross and microhematuria as well as bilirubinuria. In the face of elevated 8-iso-PGF2α, majority of the comorbid patients presented with chills, fever, anorexia, joint paints, lethargy and vomiting. It could therefore be suggested that malaria could trigger acute hemolytic crisis in sickle cell patients’. 

Review 1 Comments

Comment 1: Was the study both a retrospective and prospective Cohort study given that data was collected from people already suffering from SCD coming to the clinic for visits and people diagnosed with malaria and SCD.

Response: The study design has been changed to prospective cohort study (page 4)

Comment 2: Under study design you wrote its a cross-sectional study, could you please state the appropriate study design 

Response: The study design has been changed to prospective cohort study (page 4)

Comment 3: Given that Malaria infectivity varies seasonally, were the samples collected at same peak seasonal period or randomly? This is to minimize variations in samples parameters 

Response: Samples were collected randomly. The manuscript has been revised accordingly (page 4). 

Comment 4: In the discussion section, please elaborate more on the leukocyte ratio effect on the disease condition results 

Response: The discussion section has been improved with more information on leukocyte ratios. This information was added to the revised manuscript ‘These ratios could be diagnostic because mean proportion of eosinophils were significantly higher in comorbid state whereas in mean basophils proportion were lower in comorbid state, even though it did not reach significant level. Leukocyte ratios have been used to predict several diseases in clinical practice (48,49). But this is the first time, to the best of our knowledge, leukocyte ratios are being studied in sickle cell disease and malaria comorbidity’

Additionally, the discussion section has been revised to include more on hemolysis. The following were added ‘Critical analysis of SCD patients with malaria suggested the presence of intravascular hemolysis. This was evidenced by significantly low red blood cells with its accompanying low hemoglobin levels. Additionally, unconjugated bilirubin level was significantly elevated together with gross and microhematuria as well as bilirubinuria. In the face of elevated 8-iso-PGF2α, majority of the comorbid patients presented with chills, fever, anorexia, joint paints, lethargy and vomiting. It could therefore be suggested that malaria could trigger acute hemolytic crisis in sickle cell patients’. 

Comment 5: Was informed consent obtained before engaging the patients in the study. If yes please clearly state that and precise the age groups from which consent was obtained from

Response: the manuscript has been revised to include this statement ‘Study participants over 18 years of age provided written informed consent whereas parental assent was obtained from participants less than 18 years of age’ 

Comment 6: You stated in the manuscript that `the oxidative stress status of sickle cell patients with malaria has not been studied’. Provide more clarity on this e.g see publication below Atiku SM, Louise N, Kasozi DM. Severe oxidative stress in sickle cell disease patients with uncomplicated Plasmodium falciparum malaria in Kampala, Uganda. BMC Infect Dis. 2019;19(1):600. Published 2019 Jul 9. doi:10.1186/s12879-019-4221-y

Response: The affected statement has been revised to read ‘However, very few studies have reported oxidative stress in sickle cell patients with malaria, but none, to the best of our knowledge has been done in Ghanaian settings. Therefore, this study assessed the degree of oxidative stress in SCD patients with malaria and the associated hematological and disease presentation profile in Ghanaian patients’

Reviewer 2 comments

Comment: The authors should include in their Introduction, more about SCD pathophysiology, and about the hemolytic aspect of the disease, that is hereditary

Response: The introduction section has been improved by adding the suggested information. The following information has been added ‘HbS result from the substitution of hydrophilic glutamic acid by hydrophobic valine at the sixth position in the β-globin chain (5). Globally, 3.2 million people live with HbSS or HbSC. About 176,000 people die of HbS disease related complications every year (6). Anaemia is common in sickle patients (7) together with vaso-occlusion which frequently leads to ischemia. This cascade of events is the predominant pathophysiology responsible for acute painful vaso-occlusive crisis (8). Increased plasma viscosity occurs as a result of chronic hemolysis and reduced sickle red cell deformability due to HbS polymerization (9). These effects could be prevented or reversed by therapies that prevents HbS polymerization by allosterically modifying HbS oxygen affinity, preventing erythrocyte dehydration. Hydroxyurea, metformin and sodium butyrate are common examples (10)’ 

Comment: and also, epidemiology of malaria in the region

Response: Epidemiology of malaria has been added. Ghana, particularly, the Greater Accra region is endemic for malaria. in 2017 Ghana National Malaria Control Program report indicated that almost 48% of all Out-patient department attendants were attributable to malaria (5). Whereas in the Greater Accra region, prevalence of malaria has been reported to be 15.1% (6). In the region, malaria mostly affect children less than 15 years, males, rural and peri-urban dwellers as well as people with either no or only primary education. Additionally, unemployed and people engaging in petty trading with lower incomes are disproportionately affected (7) (page 1). 

Comment: What kind of Plasmodium is common in the region? 

Response: Over 95% of malaria cases in the region is attributable to the P. falciparum spp (8) (page 1)

Comment: The methodology is well described but will be important place about how the hospital work, and how many patients of each disease they receive. 

Response: The manuscript has been revised accordingly to include these statements ‘The Ga North Municipal Hospital is a public referral health facility that sees an average of over 200 patients a day. The hospital is a referral hospital for several smaller public and private health centres. The hospital operates an out-patient department, in-patients department, antenatal services as well as infectious and non-infectious diseases clinic. Of the average number of patients seen daily, malaria cases recorded per day is about 7 while the non-communicable section of the hospital sees about four sickle cell patients a day. Individuals in each study cohorts were randomly selected during the study period, until the pre-determined sample size was achieved’. 

Comment: The authors include Bilirubin as a renal marker, it is necessary to explain it, because there is a mistake about this data. 

Response: The table and its heading have been revised to contain all the parameters measured

Comment: About the ethics aspect, as they included patients over 10 years old, it will be important to place the parental consent and that the Declaration of Helsinki was followed. 

Response: The manuscript has been revised accordingly and it now reads ‘Study participants over 18 years of age provided written informed consent whereas parental assent was obtained from participants less than 18 years of age. Declaration of Helsinki was followed in this study.’

Comment: The authors need to place clearly that comorbidity is related to the presence of malaria and SCD. 

Response: Comorbidity has been defined in the abstract (page 2)

Comment: In the results, authors should include more specific data, and confirm data about age in the table 1 and in the text. 

Rebuttal: The results contain all the necessary information regarding table 1. It must also be noted that the ages were not presented as ranges but IQR (25th percentile – 75th percentile)

Comment: In table 2, will be important to include how many patients they found in each group of parasite density sub-range. 

Response: The number of patients in each parasitemia sub-range has been placed in brackets against the mean plasma levels of 8-iso-prostaglandin F2α

Comment: The results are well presented, but need to correct some mistake, such as, the authors referrer leukocytes as level and not count. 

Response: Throughout the manuscript, leucocyte levels have ben changed to leukocyte counts (pages 12 and 15)

Comment: The discussion needs to be rewritten the discussion, and please they need to include more about the hemolysis marker investigated at the study, there are several markers, but the authors need to justify their choice. It is important because, there is a paper that reports about the influence of hemolysis in the 8-iso-prostaglandin F2α levels, that need to be incorporated at the discussion section (Ulrike Dreiβigacker et al. Clinical Biochemistry 43 (2010) 159–167). 

Response: More information on hemolysis has been added toether with information on the relationship between hemoysis and 8-iso-PGF2α as follows ‘It is obvious that low hemoglobin observed was as a result of red cell break down which was confirmed by low red blood cell count. Profuse hemolysis observed in comorbid state was not surprising since 8-iso-PGF2α oxidative stress biomarker, which was elevated in SCD-malaria, has been associated with hemolysis (54). Enhanced metabolism of the hemoglobin yielded more bilirubin which overwhelmed the conjugation ability of the liver’.

Comment: When the authors talk about the “EBR cut-off value greater than 10 could be used to predict malaria in SCD”, there is a several important aspects that need to be incorporate such as the SCD crisis.

Response: the sentence has been revised to read ‘Therefore, EBR cut-off value greater than 10 could be used to predict malaria in SCD, provided the sickle cell patient is in stable condition without vaso-occlusive crises.’

---

## [Decision Letter · Decision Letter 1]

13 Apr 2022

PONE-D-21-30010R1Oxidative stress and associated clinical manifestations in malaria and sickle cell (HbSS) comorbidityPLOS ONE

Dear Dr. Aninagyei,

Thank you for submitting your manuscript to PLOS ONE. After careful consideration, we feel that it has merit but does not fully meet PLOS ONE’s publication criteria as it currently stands. Therefore, we invite you to submit a revised version of the manuscript that addresses the points raised during the review process. Please take into consideration the minor corrections made by reviewer 2

We look forward to receiving your revised manuscript.

Kind regards,

Gabriel Agbor

Academic Editor

PLOS ONE

Reviewers' comments:

Reviewer's Responses to Questions

**Comments to the Author**

1. If the authors have adequately addressed your comments raised in a previous round of review and you feel that this manuscript is now acceptable for publication, you may indicate that here to bypass the “Comments to the Author” section, enter your conflict of interest statement in the “Confidential to Editor” section, and submit your "Accept" recommendation.

Reviewer #1: All comments have been addressed

Reviewer #2: All comments have been addressed

2. Is the manuscript technically sound, and do the data support the conclusions?

Reviewer #1: Yes

Reviewer #2: Partly

3. Has the statistical analysis been performed appropriately and rigorously? 

Reviewer #1: Yes

Reviewer #2: Yes

4. Have the authors made all data underlying the findings in their manuscript fully available?

Reviewer #1: Yes

Reviewer #2: Yes

5. Is the manuscript presented in an intelligible fashion and written in standard English?

Reviewer #1: Yes

Reviewer #2: Yes

6. Review Comments to the Author

Reviewer #1: (No Response)

Reviewer #2: The authors carefully answered all comments made by the reviewers. The manuscript has improved considerably, with its purpose clearly stated and brings important contributions on the occurrence of malaria in individuals with sickle cell anemia. However, after reading all suggestions and recommendations, and also all results presented, this reviewer consider that the conclusion of the study, both in the abstract and in conclusion section, should be very well evaluated, once it is placed “Exogenous antioxidant supplement is suggested for sickle cell patients with malaria to neutralize the increasing levels of free radicals which are known to have deleterious consequences on cells and organs biomolecules”, since the data presented do not allow the authors to carry out the recommendation that the simple presence of the investigated marker, the isoprostane, 8-isoprostaglandin F2α (8-iso-PGF2α) serving as a basis to stimulate the use of external antioxidant agents, since the authors did not carry out functional studies that support the indication; therefore, the suggestion of this reviewer is that this recommendation should be withdrawn from the conclusion of and also from the abstract, since other markers of oxidative stress were not studied and functional studies with antioxidant therapeutic agents were not performed in this study, in a way that confirms the possibility of using these agents or specific agents in the treatment of individuals with sickle cell disease and malaria.

7. PLOS authors have the option to publish the peer review history of their article (what does this mean?). If published, this will include your full peer review and any attached files.

Reviewer #1: No

Reviewer #2: No

---

## [Author Response · Author response to Decision Letter 1]

22 Apr 2022

Reviewer #2: 

Comment: The authors carefully answered all comments made by the reviewers. The manuscript has improved considerably, with its purpose clearly stated and brings important contributions on the occurrence of malaria in individuals with sickle cell anemia. However, after reading all suggestions and recommendations, and also all results presented, this reviewer consider that the conclusion of the study, both in the abstract and in conclusion section, should be very well evaluated, once it is placed “Exogenous antioxidant supplement is suggested for sickle cell patients with malaria to neutralize the increasing levels of free radicals which are known to have deleterious consequences on cells and organs biomolecules”, since the data presented do not allow the authors to carry out the recommendation that the simple presence of the investigated marker, the isoprostane, 8-isoprostaglandin F2α (8-iso-PGF2α) serving as a basis to stimulate the use of external antioxidant agents, since the authors did not carry out functional studies that support the indication; therefore, the suggestion of this reviewer is that this recommendation should be withdrawn from the conclusion of and also from the abstract, since other markers of oxidative stress were not studied and functional studies with antioxidant therapeutic agents were not performed in this study, in a way that confirms the possibility of using these agents or specific agents in the treatment of individuals with sickle cell disease and malaria.

Response: 

We can confirm that the said recommendation has been expunged from the manuscript

---

## [Decision Letter · Decision Letter 2]

27 May 2022

Oxidative stress and associated clinical manifestations in malaria and sickle cell (HbSS) comorbidity

PONE-D-21-30010R2

Dear Dr. Enoch Aninagyei,

We’re pleased to inform you that your manuscript has been judged scientifically suitable for publication and will be formally accepted for publication once it meets all outstanding technical requirements.

Kind regards,

Gabriel Agbor

Academic Editor

PLOS ONE

Additional Editor Comments (optional):

Reviewers' comments:

Reviewer's Responses to Questions

**Comments to the Author**

1. If the authors have adequately addressed your comments raised in a previous round of review and you feel that this manuscript is now acceptable for publication, you may indicate that here to bypass the “Comments to the Author” section, enter your conflict of interest statement in the “Confidential to Editor” section, and submit your "Accept" recommendation.

Reviewer #2: All comments have been addressed

2. Is the manuscript technically sound, and do the data support the conclusions?

Reviewer #2: Yes

3. Has the statistical analysis been performed appropriately and rigorously? 

Reviewer #2: Yes

4. Have the authors made all data underlying the findings in their manuscript fully available?

Reviewer #2: Yes

5. Is the manuscript presented in an intelligible fashion and written in standard English?

Reviewer #2: Yes

6. Review Comments to the Author

Reviewer #2: (No Response)

7. PLOS authors have the option to publish the peer review history of their article (what does this mean?). If published, this will include your full peer review and any attached files.

Reviewer #2: No

---

## [Editor Report · Acceptance letter]

30 May 2022

PONE-D-21-30010R2 

Oxidative stress and associated clinical manifestations in malaria and sickle cell (HbSS) comorbidity 

Dear Dr. Aninagyei:

I'm pleased to inform you that your manuscript has been deemed suitable for publication in PLOS ONE. Congratulations! Your manuscript is now with our production department. 

Kind regards, 

on behalf of

Dr. Gabriel Agbor 

Academic Editor

PLOS ONE